# Effects of Neoadjuvant Radiotherapy on Survival in Patients with Stage IIIA-N2 Non-Small-Cell Lung Cancer Following Pneumonectomy

**DOI:** 10.3390/jcm11237188

**Published:** 2022-12-02

**Authors:** Chenghao Qu, Rongyang Li, Jingyi Han, Weiming Yue, Hui Tian

**Affiliations:** Department of Thoracic Surgery, Qilu Hospital of Shandong University, Jinan 250000, China

**Keywords:** non-small-cell lung cancer (NSCLC), pneumonectomy, chemotherapy, radiotherapy, SEER database

## Abstract

Background: Pneumonectomy is a drastic but sometimes inevitable treatment option for patients with non-small-cell lung cancer (NSCLC) to improve their chances for long-term survival. However, the optimal adjuvant radiotherapy used for patients with N2 NSCLC following pneumonectomy remains unclear in the literature. Methods: T1-4N0-2M0 NSCLC patients registered in the Surveillance, Epidemiology, and End Results database were retrospectively analyzed. Propensity score matching was applied to balance the assignment of patients. Cox proportional hazards models and Kaplan–Meier analyses were used to identify the factors related to overall survival rates. Restricted cubic splines were used to detect the possible nonlinear dependency of the relationship between the risk of survival and age. Results: A total of 4308 NSCLC patients were enrolled in this study. In N2 patients, the long-term outcome of the chemotherapy and postoperative radiotherapy groups was the worst (*p* = 0.014). Subgroup analyses showed that the influence of age on survival outcome was confined to patients who received chemotherapy and neoadjuvant radiotherapy (*p* = 0.004). Meanwhile, patients >65 years of age who received chemotherapy and neoadjuvant radiotherapy had significantly worse prognoses than those in the chemotherapy group (*p* = 0.005). Conclusions: Our results show that neoadjuvant radiotherapy may have potential benefits in patients aged ≤ 65 years who are scheduled for pneumonectomy, but not in elderly patients.

## 1. Introduction

Professor Graham performed a left pneumonectomy in 1933. This event laid the foundation for using this type of procedure to treat lung cancer [1]. However, pneumonectomy is associated with early adverse outcomes and high mortality rates [2,3]. Sleeve-lobectomy and other surgical approaches can reduce the need for pneumonectomy to be performed [4]. Nevertheless, in certain cases, pneumonectomy is an inevitable procedure to achieve long-term survival outcomes.

Multimodal treatment, including chemotherapy (CT) and radiotherapy (RT), followed by surgery is also recommended for patients with potentially resectable cN2 non-small-cell lung cancer (NSCLC) [5,6]. Perioperative adjuvant CT has shown significant survival benefits in pN2 NSCLC patients [7,8,9]. However, an optimal form of adjuvant radiation therapy in patients with cN2 NSCLC has not yet been established because of controversial results [10,11].

As an additional local treatment, neoadjuvant radiotherapy (NART) and postoperative radiotherapy (PORT) have significant effects on prolonged survival rates by reducing the local recurrence rate in patients with cN2 NSCLC following surgery [12,13,14]. However, some studies have determined that significant adverse effects of PORT may offset the advantage of survival when patients receive adjuvant CT [15]. In particular, PORT may not improve survival rates in elderly patients over 75 years of age with N2-stage NSCLC [13,16]. To address this issue, we aimed to investigate the benefits of perioperative adjuvant radiotherapy following pneumonectomy in patients with NSCLC. We present this article in accordance with the STROBE reporting checklist.

## 2. Materials and Methods

### 2.1. Data Source

The data files for this research were retrieved from the Surveillance, Epidemiology, and End Results (SEER) database. This population study used the SEER-18 dataset with SEER* Stat v. 8.3.8 software. The data released by the SEER database did not require informed patient consent. This study was approved by the Committee for Ethical Review of Research in Qilu Hospital. Ethical approval was not required as it was the secondary use of collected data obtained from the SEER database.

### 2.2. Patients

In this retrospective study, we used the Incidence—SEER 18 Regs Custom Data (with additional treatment fields) to identify NSCLC patients who were diagnosed via pathological examination and underwent radical pneumonectomy and lymph node dissection between January 2004 and December 2016. There were 4447 patients diagnosed with T1-4N0-2M0. Patients who received RT alone and those with an unclear RT time were excluded from our study. The following data were extracted from the database: age, sex, race, laterality of surgery, pathological type, grade, TNM stage, marital status, surgery methods, use of adjuvant therapy, and type of adjuvant therapy. The 8th edition of the TNM staging system was used in the present study.

### 2.3. Statistical Analysis

The demographic and clinical characteristics of patients were compared among different groups using one-way analysis of variance for continuous variables and Pearson’s chi-squared test for categorical variables. Cox proportional hazards models were used to identify the risk factors associated with overall survival (OS). Kaplan–Meier analyses were used to generate survival curves, and the log-rank test was applied to analyze the differences among the curves. To balance the assignment of patients, we used 1:1 propensity score matched (PSM) analyses according to the adjuvant treatment methods applied by the patients. The propensity score model included age, sex, surgery laterality, pathological type, grade, TNM stage, and marital status. The caliper value was set to 0.2. Standardized difference (SD) and absolute standardized mean difference (ASMD) were used to assess the adequacy of balance between the cohort prior to and following the matching procedure. An ASMD ≤ 10% indicated adequate balance [17]. Restricted cubic splines (RCSs) were used to detect the possible nonlinear dependency of the relationship between the risk of death and age levels, using four knots at prespecified locations, according to the percentiles of the distribution for 50, 60, 70, and 80 years of age. Statistical analyses were conducted using SPSS version 25 (IBM, New York, NY, USA) and R (version 4.0.3; Vienna, Austria). Statistical significance was defined as a two-sided *p*-value < 0.05.

## 3. Results

A total of 4308 patients were enrolled in the present study. The average of follow-up time was 41.6 months. The patients were divided into three groups based on the treatment strategy: surgery only (Surg, *n* = 2156), surgery with CT (including pre- and postoperative CT) (CT, *n* = 1380), and surgery with chemoradiotherapy (CRT) (including pre- and postoperative CRT) (CRT, *n* = 772). There were no significant differences between the three groups in sex, race, pathological type, or extended pneumonectomy. The mean age was the lowest in patients who underwent surgery only (Table 1).

Cox proportional hazards models included age, sex, race, laterality, pathological type, grade, T stage, N stage, being the only cancer present, marital status, surgery, and treatment. Table 2 presents the results of univariate and multivariate analyses of the OS.

Furthermore, the patients were also divided into N0 (*n* = 1656), N1 (*n* = 1675), and N2 (*n* = 977) groups according to their lymph node status. The Kaplan–Meier analysis showed that there were significant differences in survival in the N0 group between the patients who underwent surgery with CT and those in the other two groups (*p* < 0.001) (Figure 1A).

In the N1 and N2 groups, patients who received CT and CRT had a better prognosis than those who only received surgical therapy (*p* < 0.001) (Figure 1B,C). In particular, the OS in patients who underwent surgery with CT was significantly better than that in patients who underwent surgery with CRT in the N1 group (*p* < 0.001) (Figure 1B), while the OS was comparable in the respective subgroup analysis in the N2 group (*p* = 0.614) (Figure 1C).

From the N2 group (*n* = 691), we excluded 44 patients whose RT sequences were not clear. The remaining 647 patients were divided into CT and CRT groups. To investigate the different prognoses of CT and CRT, we performed PSM analyses, identified 440 patients (Table 3), and divided them into three subgroups: CT (*n* = 220), CT + NART (*n* = 93), and CT + PORT (*n* = 127). CT group was set as the control group. Following the PSM, patients with CT + PORT had the worst prognosis (CT vs. CT + PORT: *p* = 0.014) (Figure 2).

Research has shown that older patients may not benefit from adjuvant therapy, especially from PORT [16]. We used RCS to investigate the influence of age further and observed that it was a prognostic factor in N2 patients (*p* = 0.016). However, to our surprise, the influence of age was confined only to the CT + NART group following subgroup analysis (Figure 3) (CT: *p* = 0.617; CT + NART: *p* = 0.004; CT + PORT: *p* = 0.629). We divided patients after the PSM into two groups using a cut-off age of 65 years. The prognosis in the CT group was significantly better than that in the CT + PORT group in patients below 65 years of age (*p* = 0.028). However, there was no significant difference in the prognosis between the two methods used in patients older than 65 years of age (*p* = 0.482) (Figure 4A, B).

An obvious difference in prognosis between age subgroups was observed in the CT + NART group. In patients aged ≤ 65 years, the OS in the CT + NART group was better than that in the CT group, although this difference was not statistically significant (*p* = 0.443) (Figure 4A). However, the OS was significantly worse in patients over 65 years of age (*p* = 0.005) (Figure 4B).

## 4. Discussion

This study demonstrated that for N2-stage NSCLC patients undergoing pneumonectomy, PORT and NART need to be cautiously conducted. NART may have some clinical value for N2 NSCLC patients aged ≤ 65 years who are ready to undergo pneumonectomy. For N0-N1-stage NSCLC patients undergoing pneumonectomy, RT is not associated with a better prognosis.

In this study, 4308 patients from the SEER database were divided into three groups based on different treatments. As predicted, patients who received adjuvant therapy were significantly younger than those who underwent surgery alone. This may be because the expected potential adverse outcomes associated with CRT in elderly patients resulted in their doctors avoiding offering this treatment. Previous studies have shown that patients undergoing pneumonectomy who received different types of adjuvant therapy during the perioperative period may have different outcomes.

The role of neoadjuvant or postoperative CRT in patients with NSCLC remains controversial [14,18,19]. Adjuvant CT is the standard treatment for patients with resected node-positive NSCLC. In general, adjuvant CT has little impact on the mortality rates in pneumonectomy patients [20]. A recent study showed that multimodal treatment based on RT was considered more frequently in patients with an increasing extent of mediastinal nodal disease [21]. However, toxicity is considerable when patients receive a combination treatment of RT and CT [22].

Generally, elderly patients tend to have more comorbidities and an inferior physical status than younger patients. The study proved that elderly patients who are fit to receive CRT have a better prognosis [23]. However, it is still difficult to identify patients who are suitable to be treated with CRT, especially among those undergoing pneumonectomy. A previous study showed that patients older than 75 years of age received no benefits from neoadjuvant RT [24]. By using RCS, our study proved that age remained a prognostic factor in the CT + NART group. We set three treatment groups to confirm that age had a significant effect only on the CT + NART group, but not in other treatment groups. In our study, patients aged ≤65 years received significant benefits from NART and presented the best long-term outcomes. However, patients older than 65 years of age receiving CT + NART had the worst OS outcome. The result may have been due to the fact that patients with pneumonectomy had a higher risk of complications compared with those receiving lobectomy. Older adults are physically weaker and more vulnerable to the side effects of CRT. Regardless of the reasons, this result may indicate that CRT should be performed more cautiously on elderly patients.

The role of RT in pneumonectomy IIIA-N2 NSCLC has not been defined in the literature. For patients with pIIIA-N2 who undergo pneumonectomy, it is necessary to explore who might achieve the greatest benefits from receiving RT. Our study showed that elderly patients (>65 years) may present adverse long-term outcomes when receiving neoadjuvant RT. This means that the effect of age is more significant in patients receiving neoadjuvant RT than in those receiving PORT.

Based on our study, the choice of RT should be evaluated with caution for N2 NSCLC patients who undergo pneumonectomy and adjuvant CT. In resectable cN2 patients, NART is sometimes used for downstaging to achieve better operational preparation. Previous research has shown that patients with IIIB NSCLC receiving CT and RT followed by surgery have a favorable OS outcome [22]. However, our results show that this choice may lead to a worse prognosis in elderly patients (>65 years) who are ready to undergo pneumonectomy.

In reality, some patients with N2 disease undergo NART for a chance to undergo pneumonectomy. However, in many cases, some patients with N2 disease undergo NART to achieve better preparation for the surgery. Our research results show that a decision to receive NART aiming to achieve better preparation for pneumonectomy may not be suitable for elderly IIIA-N2 patients (>65 years). However, the definition of resectability of N2 disease is too complex, which makes it difficult to make the best treatment decisions. Neoadjuvant treatment may also lead to some risks, including mediastinal soft tissue fibrosis, potentially complicating subsequent hilar and mediastinal dissections, which may make surgery more difficult [25]. Therefore, trimodal treatment decisions are best made by an experienced multidisciplinary team. Further investigations of RT should be conducted with prospective randomized studies of patients with N2 NSCLC undergoing pneumonectomy.

Recent studies based on the SEER database showed that PORT alone and PORT combined with postoperative chemotherapy could prolong the OS in patients with N2 NSCLC [12,14,26]. This year, clinical trial Lung ART presented a result that proved that PORT was not associated with an increased disease-free survival outcome compared with no PORT [27]. This result is similar to a recent randomized controlled trial, PORT-C, which showed that PORT with adjuvant CT did not improve disease-free survival or OS outcomes in IIIA-N2 NSCLC patients following a complete resection [28]. However, our research proved that in N2 NSCLC patients undergoing pneumonectomy, PORT combined with postoperative chemotherapy may be associated with a poor prognosis. This result may not be robust enough. However, this result proves PORT should be carefully implemented in pneumonectomy patients. Further clinical trials should be conducted to accurately identify pN2 pneumonectomy patients who can benefit from RT.

Immunotherapy has recently brought a paradigm shift in the treatment of NSCLC [29,30,31]. Neoadjuvant chemo-immunotherapy resulted in significantly longer event-free survival and a higher percentage of patients with a pathological complete response than CT alone in resectable NSCLC [32,33]. Meanwhile, the addition of immunotherapy did not increase the incidence of adverse events or impede the feasibility of surgery [33,34]. An important issue is that the surgical procedure may be more technically challenging because of hilar inflammation and fibrosis. However, according to the published data, it seems that surgery is safe and possible following use of neoadjuvant immunotherapy [35]. Based on the results reported to date, the combination of immunotherapy and CT appears to be a promising strategy in pneumonectomy. But more robust data are needed to definitively establish the most appropriate treatment regimen for this combined approach.

We acknowledge that there were several limitations to this study. As a retrospective investigation, treatment regimens may be influenced by many factors. Meanwhile, the lack of detailed information about targeted therapy and immunotherapy was another limitation. Due to the records of the SEER database lacking recurrence information, we only set the OS as the main outcome. However, since the number of pneumonectomy patients was generally small, we believe that this study, including 4308 patients, is of importance, despite the limitations mentioned above.

## 5. Conclusions

In conclusion, For N2-stage NSCLC patients undergoing pneumonectomy, PORT and NART need be conducted cautiously. NART may have some clinical value for patients aged ≤ 65 years who are ready to undergo pneumonectomy. For N0-N1-stage NSCLC patients undergoing pneumonectomy, RT is not associated with a better prognosis. The combination of immunotherapy and CT appears to be a promising strategy in pneumonectomy.

## Figures and Tables

**Figure 1 jcm-11-07188-f001:**
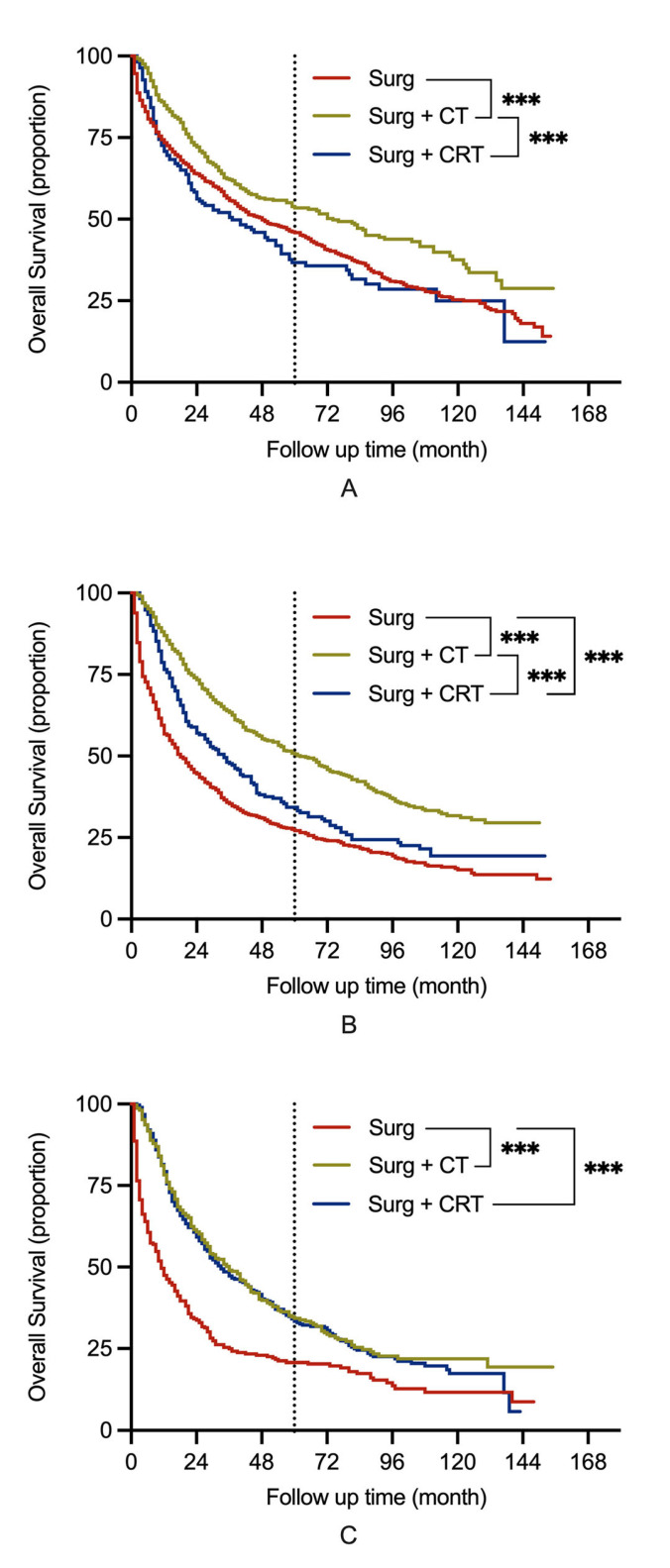
Overall survival in NSCLC patients with N0-2 stage undergoing pneumonectomy. (**A**) N0 stage; (**B**) N1 stage; (**C**) N2 stage. Note: Surg, surgery; CT, chemotherapy; CRT, chemoradiotherapy; NSCLC, non-small cell lung cancer; vs., versus; *** Indicates significant difference at *p* < 0.001.

**Figure 2 jcm-11-07188-f002:**
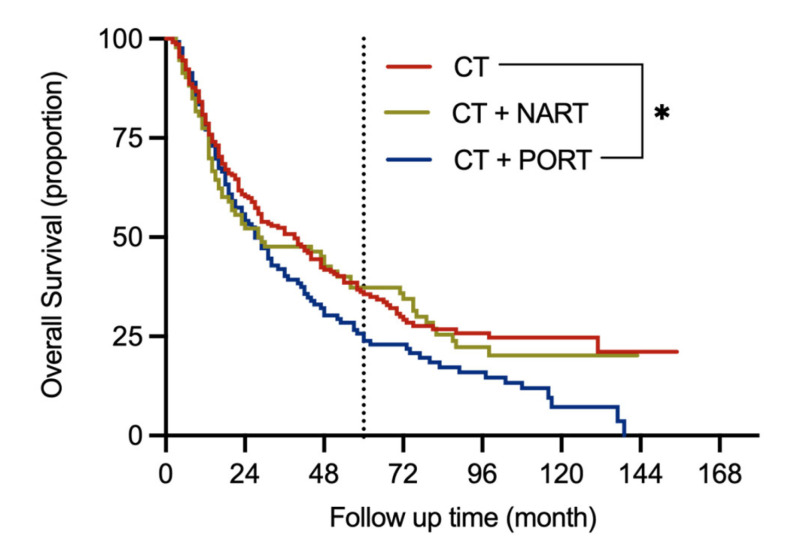
Overall survival in different treatment groups of NSCLC patients with N2 stage after propensity score matching. Note: CT, chemotherapy; NART, neoadjuvant radiotherapy; NSCLC, non-small cell lung cancer; PORT, postoperative radiotherapy; * Indicates significant difference at *p* < 0.05.

**Figure 3 jcm-11-07188-f003:**
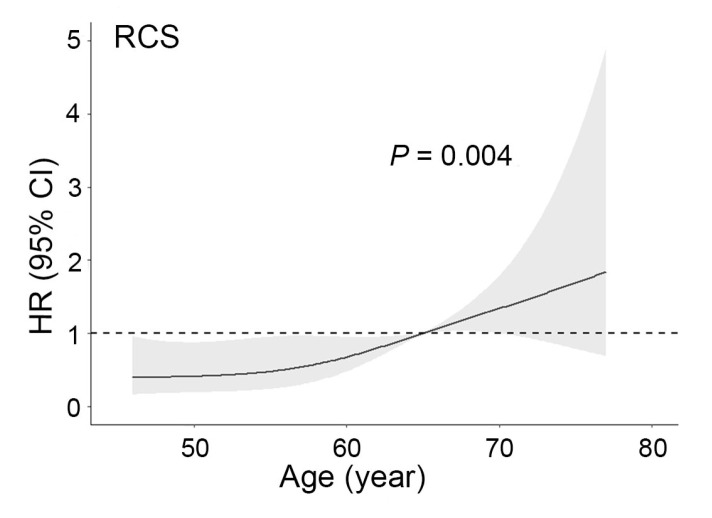
Nonlinear dependence between the risk of death and age in NSCLC patients with N2 stage who received chemotherapy and neoadjuvant radiotherapy. Note: NSCLC, non-small cell lung cancer; RCS, restricted cubic splines; HR, hazard ratio; CI, confidence interval.

**Figure 4 jcm-11-07188-f004:**
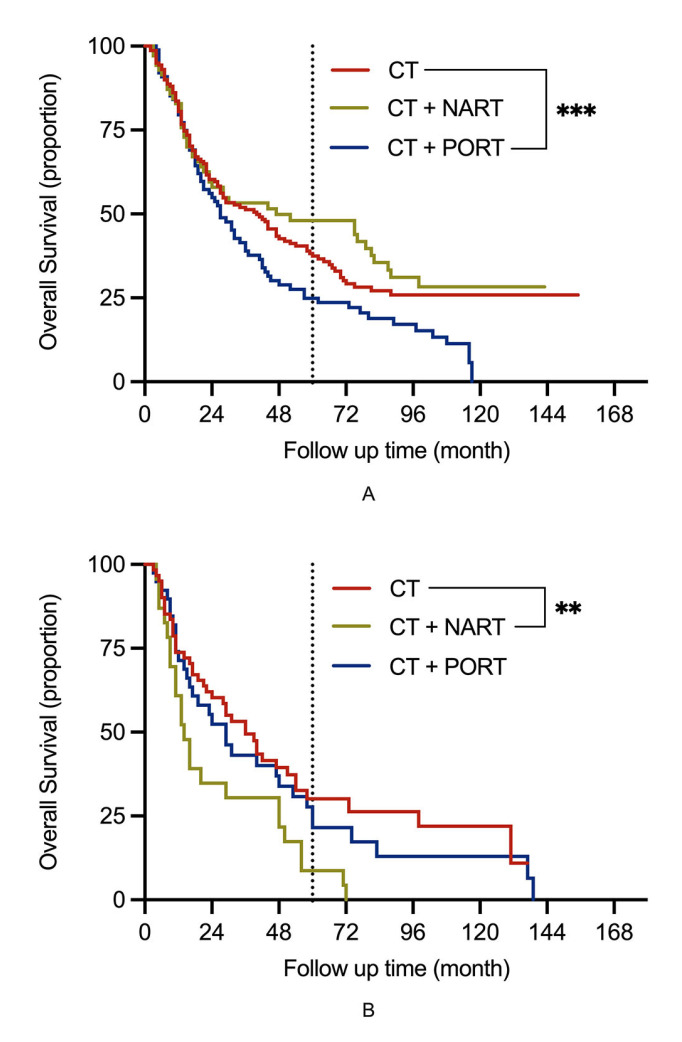
Overall survival in NSCLC patients with N2 stage after propensity score matching. (**A**) in patients below 65 years (≤65); (**B**) in patients above 65 years (>65). Note: CT, chemotherapy; NSCLC, non-small cell lung cancer; NART, neoadjuvant radiotherapy; PORT, postoperative radiotherapy; ** Indicates significant difference at *p* < 0.01; *** Indicates significant difference at *p* < 0.001.

**Table 1 jcm-11-07188-t001:** Baseline characteristics of patients who underwent pneumonectomy for non–small-cell lung cancer.

Characteristics	Treatment (*N* = 4308)			*p*
	Surg *N* = 2156 (%)	CT *N* = 1380 (%)	CRT *N* = 772 (%)	
Age (Year)	65.56	61.49	59.73	<0.001 ^†,^*
Sex				
Male	1433 (66.5)	877 (63.6)	518 (67.1)	0.132
Female	723 (33.5)	503 (36.4)	254 (32.9)	
Race				
Black	173 (8.0)	111 (8.0)	66 (8.5)	0.428
White	1877 (87.1)	1181 (85.6)	666 (86.3)	
Others	106 (4.9)	88 (6.4)	40 (5.2)	
Laterality				
Left	1202 (55.8)	831 (60.2)	465 (60.2)	0.012 *
Right	954 (44.2)	549 (39.8)	307 (39.8)	
Pathological type				
SCC	1127 (52.3)	722 (52.3)	410 (53.1)	0.787
ADC	900 (41.7)	565 (40.9)	320 (41.5)	
Others	129 (6.0)	93 (6.7)	42 (5.4)	
Grade				
I/II	987 (45.8)	566 (41.0)	276 (35.8)	<0.001 *
III	1038 (48.1)	748 (54.2)	390 (50.5)	
Unknown	131 (6.1)	66 (4.8)	106 (13.7)	
T				
1	348 (16.1)	113 (8.2)	43 (5.6)	<0.001 *
2	1250 (58.0)	836 (60.6)	356 (46.1)	
3	239 (11.1)	197 (14.3)	150 (19.4)	
4	319 (14.8)	234 (17.0)	223 (28.9)	
N				
0	1119 (51.9)	372 (27.0)	165 (21.4)	<0.001 *
1	751 (34.8)	694 (50.3)	230 (29.8)	
2	286 (13.3)	314 (22.8)	377 (48.8)	
Stage				
I	902 (41.8)	253 (18.3)	69 (8.9)	<0.001 *
II	627 (29.1)	546 (39.6)	149 (19.3)	
III	627 (29.1)	581 (42.1)	554 (71.8)	
The only cancer				
Yes	1429 (66.3)	985 (71.4)	590 (76.4)	<0.001 *
No	727 (33.7)	395 (28.6)	182 (23.6)	
Marital status				
Single	265 (12.3)	174 (12.6)	79 (10.2)	<0.001 *
Married	1287 (59.7)	876 (63.5)	527 (68.3)	
Divorced	285 (13.2)	185 (13.4)	100 (13.0)	
Widowed	257 (11.9)	10 (7.9)	46 (6.0)	
Unknown	62 (2.9)	36 (2.6)	20 (2.6)	
Surgery				
Pneumonectomy	2088 (96.8)	1325 (96.0)	741 (96.0)	0.330
Extended pneumonectomy	68 (3.2)	55 (4.0)	31 (4.0)	

Surg: surgery only; CT: surgery with chemotherapy; CRT: surgery with chemoradiotherapy. Abbreviations: SCC, squamous cell carcinoma; ADC, adenocarcinoma. Footnote: * Indicates significant difference at *p* < 0.05. ^†^ *p*-value obtained from one-way ANOVA. Data are presented as *n* (%) or means.

**Table 2 jcm-11-07188-t002:** Cox proportional hazard regression model for overall survival in patients who underwent pneumonectomy for non–small-cell lung cancer.

Characteristics	Univariate Analysis			Multivariate Analysis		
	HR	95.0% CI	*p*	HR	95.0% CI	*p*
Age	1.026	1.022–1.029	<0.001 *	1.026	1.022–1.030	<0.001 *
Sex						
Male	Ref.			Ref.		
Female	0.775	0.716–0.839	<0.001 *	0.750	0.689–0.816	<0.001 *
Race						
Black	Ref.					
White	1.060	0.923–1.216	0.410			
Others	0.927	0.750–1.146	0.486			
Laterality						
Left	Ref.			Ref.		
Right	1.183	1.098–1.273	<0.001*	1.220	1.132–1.314	<0.001 *
Pathological type						
SCC	Ref.			Ref.		
ADC	1.002	0.928–1.082	0.950	1.096	1.011–1.189	0.027 *
Others	1.200	1.033–1.394	0.017*	1.140	0.977–1.329	0.097
Grade						
I/II	Ref.			Ref.		
III	1.222	1.132–1.319	<0.001 *	1.190	1.099–1.288	<0.001 *
Unknown	1.097	0.941–1.277	0.236	1.044	0.894–1.220	0.585
T						
1	Ref.			Ref.		
2	1.101	0.977–1.242	0.116	1.082	0.957–1.225	0.208
3	1.266	1.093–1.468	0.002 *	1.322	1.134–1.541	<0.001 *
4	1.388	1.208–1.594	<0.001 *	1.352	1.171–1.560	<0.001 *
N						
0	Ref.			Ref.		
1	1.198	1.100–1.304	<0.001 *	1.402	1.282–1.532	<0.001 *
2	1.421	1.291–1.564	<0.001 *	1.798	1.618–1.998	<0.001 *
The only cancer						
Yes	Ref.					
No	0.976	0.903–1.056	0.550			
Marital status						
Single	Ref.			Ref.		
Married	0.924	0.823–1.037	0.181	0.802	0.713–0.902	<0.001 *
Divorced	0.962	0.832–1.113	0.606	0.881	0.760–1.020	0.089
Widowed	1.213	1.039–1.415	0.014 *	0.950	0.807–1.120	0.544
Unknown	0.794	0.609–1.035	0.088	0.782	0.600–1.021	0.070
Surgery						
Pneumonectomy	Ref.					
Extended pneumonectomy	1.375	1.139–1.659	0.001 *	1.421	1.175–1.718	<0.001 *
Treatment						
Surgery	Ref.			Ref.		
Chemotherapy	0.622	0.570–0.678	<0.001 *	0.573	0.523–0.628	<0.001 *
Chemoradiotherapy	0.863	0.782–0.953	0.003	0.733	0.656–0.819	<0.001 *

Abbreviations: HR, Hazard Ratio; CI, confidence interval; Ref., Reference. Note: * Indicates significant difference at *p* < 0.05.

**Table 3 jcm-11-07188-t003:** Patient baseline characteristics by postoperative treatment before and after propensity score matching.

Characteristics	Before Matching (*N* = 647)				After Matching (*N* = 440)				
	CT *N* = 302 (%)	CRT *N* = 345 (%)	*p*	SD (%)	CT *N* = 220 (%)	CRT *N* = 220 (%)	*p*	SD (%)	ASMD (%)
Age (year)	60.57	59.02	0.04 ^†,^*	16.2	59.11	60.75	0.060 ^†,^*	17.9	1.7
Sex			0.369	7.1			0.920	1.0	0.6
Male	184 (60.9)	222 (64.3)			143 (65.0)	144 (65.5)			
Female	118 (39.1)	123 (35.7)			77 (35.0)	76 (34.5)			
Laterality			0.371	7.1			0.772	2.8	0.7
Left	176 (58.3)	189 (54.8)			128 (58.2)	125 (56.8)			
Right	126 (41.7)	156 (45.2)			92 (41.8)	95 (43.2)			
Pathological type			0.088	17.3			0.794	6.5	1.9
SCC	147 (48.7)	166 (48.1)			110 (50.0)	104 (47.3)			
ADC	131 (43.4)	165 (47.8)			101 (45.9)	108 (49.1)			
Others	24 (7.9)	14 (4.1)			9 (4.1)	8 (8)			
Grade			<0.001 *	33.6			0.972	2.3	0.5
I/II	122 (40.4)	127 (36.8)			91 (41.4)	91(41.4)			
III	164 (54.3)	165 (47.8)			120 (54.5)	119 (54.1)			
Unknown	16 (5.3)	53 (15.4)			9 (4.1)	10 (4.5)			
T			0.335	14.6			0.981	4.0	1.3
1	24 (7.9)	24 (7.0)			15 (6.8)	16 (7.3)			
2	179 (59.3)	184 (53.3)			125 (56.8)	121 (55.0)			
3	34 (11.3)	49 (14.2)			28 (12.7)	30 (13.6)			
4	65 (21.5)	88 (25.5)			52 (23.6)	53 (24.1)			
Marital status			0.144	11.5			0.614	4.8	0.7
Married	191 (63.2)	237 (68.7)			143 (65.0)	148 (67.3)			
Others	111 (36.8)	108 (31.3)			77 (35.0)	72 (32.7)			

CT: surgery with chemotherapy; CRT: surgery with chemoradiotherapy. Abbreviations: SD, standardized difference; ASMD, absolute standardized mean difference; SCC, squamous cell carcinoma; ADC, adenocarcinoma. Note: * Indicates significant difference at *p* < 0.05. ^†^ *p*-value obtained from one-way ANOVA. ASMD ≤ 10% indicates adequate balance. Data are presented as *n* (%) or means.

## Data Availability

The datasets analyzed during the current study are available in the SEER database: http://seer.cancer.gov (accessed on 15 August 2021).

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
