# Peer review of "Effects of Neoadjuvant Radiotherapy on Survival in Patients with Stage IIIA-N2 Non-Small-Cell Lung Cancer Following Pneumonectomy"

_jcm, 2022, doi:10.3390/jcm11237188_

Round 1

Reviewer 1 Report

Thank you for the opportunity to review the manuscript: "Effects of neo-adjuvant radiotherapy on survival in patients with stage IIIA-N2 non-small cell lung cancer after pneumonectomy", written by Dr. Chenghao Qu and coauthors.

The manuscript in the field of major pulmonary surgery and is very informative. I have major comment regarding systematization of the manuscript that  will change, in my opinion, the results and conclusions:

This is a retrospective study on 4308 pnemonectomies. The study examined 3 groups of patients:

a. Those that have undergone only surgical treatment (2156 patients),

b. Those after chemotherapy (1380),

c. And those after chemo-radiation ( 772).

From the beginning of the study it is not completely understood, what was the type & the order of the treatment in the patients that underwent pneumonectomy:

a. have the surgical  group  patients received any oncologic (and what kind) pre-operative or postoperative treatment?

b.  have the chemotherapy group received chemotherapy before or after the surgery?

c.  in the chemo-radiation group, the patients received the treatment before of after surgery?

it is difficult to understood and because of this, it is difficult understand the conclusions of the study (Results, Table 2 (in the bottom), Figures and Graphs, Discussion).

In addition, in the Discussion we see  subgroups of patients (NART = neo-adjuvant chemotherapy and PORT= post-operative chemotherapy), that are the subgroups  of the above mentioned 3 groups that were not clear beforehand.

This is a retrospective study on patients T1-4N0M0 NSCLC that underwent major surgery and for future recommendations it is very important to reorganize  the study, I recommend the following  plan, if the authors are interested, to present the results on this big series of patients, by forming only  2 groups:

Group I: patients which have received any oncologic pre-operative treatment (neo-adjuvant chemotherapy or chemo-radiaotherapy), without additional division into subgroups,  an have undergone adjuvant pneumonectomy afterwards (with or without additional postoperative oncologic (any) treatment).

Group II: patients that have underwent pneumonectomy without preoperative oncologic treatment (neoadjuvant chemotherapy or chemoradiaotherapy) (with or without additional postoperative adjuvant oncologic (any) treatment).

It is obvious that all the results and conclusions will be changed. I also recommend dividing the morbidity, mortality and survival results (and other characteristics) according to 2 above mentioned groups:

In my opinion in this form the manuscript will add significantly to the understanding of major lung surgery (pneumonectomy) in the era of modern oncologic therapies.

Thank you very much.

Reviewer 2 Report

The authors present a retrospective analysis using the SEER database of patients treated with pneumonectomy with or without additional therapies. It seems like the goal is to see which populations may benefit from these additional therapies. As it is currently presented the study has severe limitations, lacks focus, and will require extensive revisions.

Major comments:

-The manuscript is entitled "Effects of neoadjuvant radiotherapy on survival in patients with stage IIIA-N2 non-small cell lung cancer after pneumonectomy" but yet includes all stages and patients who received PORT or no RT. If I'm honest, the potential novelty is limited to the population specified in the title so I agree with that. I just think the analysis and inclusion should be refined. 

-The role of PORT has been well investigated. This manuscript does not contribute anything to the PORT literature. I recommend it be removed. This is especially true in patients treated with pneumonectomy as they would be even more susceptible to RT toxicity. 

-The role of pneumonectomy is increasingly limited. It is not clear why the authors limited their population to pneumonectomy, and the authors should consider expanding surgical inclusion criteria as more limited surgeries are increasingly the standard of care. Alternatively, at minimum the years the patients were treated should be detailed more precisely (not just range) so we know if the patients are representative of more modern clinical practice. In my mind, the main question of interest is who benefits from the addition of neoadjuvant RT to chemo before surgery, regardless of type of surgery (or even stratified by type of surgery).

-The role of radiation in resectable NSCLC is also increasingly limited. The main intent of neoadjuvant treatment is converting unresectable patients to resectable (and hopefully limit role of pneumonectomy as well). This is primarily limited to N2 patients. Including patients with N0-1 disease does not add to the literature. These patients should not be getting RT if complete resection is possible. So analysis should be done limited to N2 patients treated with neoadjuvant therapy and maybe surgery alone. 

-Based on the above point, the discussion should focus more on other neoadjuvant approaches, including immunotherapy based on Checkmate 816.

-The main interest of this study is who may benefit from neoadjuvant RT. The age analysis is interesting. The discussion should focus on this and similar results. 

-Due to the above comments, Figure 1 is not interesting as it does not just look at patients treated with or without neoadjuvant RT. Figures 2 and 4 should also be redone with updated inclusion criteria, without including the PORT arm. Figure 3 should also be done with updated inclusion criteria. 

-I suggest a patient selection schema so it is clear which patients are included in each analysis. 

Minor Comments:

-Propensity score matching does not eliminate confounding as the abstract says. This should be rephrased.

-If discussing PORT, both LUNG-ART and PORT-C should be cited

-The table includes Grade IV. Is this meant to be stage? Grade should range from 1-3.

Round 2

Reviewer 2 Report

I do not think my comments have been adequately addressed. The title still suggests a focus on neoadjuvant RT in stage IIIA-N2 but PORT and non stage IIIA-N2 are included. The question the title is addresssing has novelty. Questions outside that inclusion have randomized clinical trials that do not need to be investigated with retrospective SEER studies and all of their biases. 

The results regarding PORT are entirely unsurprising. If DFS is not improved with PORT following lobectomy, I certainly wouldn't recommend it after a more invasive pneumonectomy, where the benefit is almost certainly less. Therefore, I continue to recommend removing discussion of PORT. This will be helpful because the article currently lacks focus and is hard for the reader to follow. Omitting the discussion of PORT, which now has two randomized trials against it, will allow you to focus on  the more interesting components (neoadjuvant therapies and relationship with age). I do not think keeping this in only to show a lack of interaction with age is novel enough to overcome how it detracts from the overall focus. 

I understand that the SEER database does not include neoadjuvant regimens. Nonetheless, they can still be included in the discussion as they are highly relevant to the next steps of the field. 

Author Response

Thank you very much for your review.

We are very sorry for that we did not address your comments adequately in the first revasion.

After we reviewed the discussion again, we very much agree with your comment that “This will be helpful because the article currently lacks focus and is hard for the reader to follow.” We had an in-depth discussion and made major revision.

  1. In this vision of manuscript, we removing discussion of PORT and we reframe the discussion to highlight the salient interest point more clearly.

(Page 8, line 173-181, line 196, line 200, line 203, line 220-232)

  1. We followed your suggestion of neoadjuvant regimens. In the end of discussion, we added neoadjuvant Immunotherapy because we think the combination of immunotherapy and chemotherapy appears to be a promising strategy in pneumonectomy. We hope that you agree with us.

(Page 10, line 233-244, line 256-257)

There is no doubt that your suggestions have played a very important role in improving the quality of the manuscript especially the discussion!

Thank you.
